# Integrated Gel Electrophoresis and Mass Spectrometry Approach for Detecting and Quantifying Extraneous Milk in Protected Designation of Origin Buffalo Mozzarella Cheese

**DOI:** 10.3390/foods14071193

**Published:** 2025-03-28

**Authors:** Sabrina De Pascale, Giuseppina Garro, Silvia Ines Pellicano, Andrea Scaloni, Stefania Carpino, Simonetta Caira, Francesco Addeo

**Affiliations:** 1Proteomics, Metabolomics & Mass Spectrometry Laboratory, Institute for the Animal Production System in the Mediterranean Environment, National Research Council, 80055 Portici, Italy; sabrinadepascale@cnr.it (S.D.P.); andrea.scaloni@cnr.it (A.S.); 2Department of Agriculture, University of Naples “Federico II”, 80055 Portici, Italy; giuseppina.garro@unina.it (G.G.); doglie42@gmail.com (F.A.); 3Central Inspectorate for Fraud Repression and Quality Protection of the Agrifood Products and Food, Ministry of Agricultural, Food and Forestry Policies, 06128 Perugia, Italy; s.pellicano@masaf.gov.it; 4Central Inspectorate for Fraud Repression and Quality Protection of the Agrifood Products and Food, Ministry of Agricultural, Food and Forestry Policies, 00187 Roma, Italy; s.carpino@masaf.gov.it

**Keywords:** buffalo milk adulteration, gel electrophoresis, MALDI-TOF-MS, nano-LC-ESI-MS/MS, peptidomics, proteomics, food forensic science

## Abstract

Ensuring the authenticity of Mozzarella di Bufala Campana (MdBC), a Protected Designation of Origin (PDO) cheese, is essential for regulatory enforcement and consumer protection. This study evaluates a multi-technology analytical platform developed to detect adulteration due to the addition of non-buffalo milk or non-PDO buffalo milk in PDO dairy buffalo products. Peripheral laboratories use gel electrophoresis combined with polyclonal antipeptide antibodies for initial screening, enabling the detection of foreign caseins, including those originating outside the PDO-designated regions. For more precise identification, Matrix-Assisted Laser Desorption Ionization Time of Flight Mass Spectrometry (MALDI-TOF-MS) differentiates species by detecting proteotypic peptides. In cases requiring confirmation, nano-liquid chromatography coupled to electrospray tandem mass spectrometry (nano-LC-ESI-MS/MS) is used in central state laboratories for the highly sensitive detection of extraneous milk proteins in PDO buffalo MdBC cheese. On the other hand, analysis of the pH 4.6 soluble fraction from buffalo blue cheese identified 2828 buffalo-derived peptides and several bovine specific peptides, confirming milk adulteration. Despite a lower detection extent in the pH 4.6 insoluble fraction following tryptic hydrolysis, the presence of bovine peptides was still sufficient to verify fraud. This integrated proteomic approach, which combines electrophoresis and mass spectrometry technologies, significantly improves milk adulteration detection, providing a robust tool to face increasingly sophisticated fraudulent practices.

## 1. Introduction

Mozzarella di Bufala Campana (MdBC) is an Italian cheese recognized with Protected Designation of Origin (PDO) status and classified as pasta filata. It is produced exclusively in specific regions of Italy, including Puglia, Campania, Lazio, and Molise, using milk sourced solely from Mediterranean buffaloes raised in these territories. However, the increasing global demand and seasonal fluctuations in milk availability have led to fraudulent practices, such as adding non-Mediterranean buffalo or cow milk to PDO cheese.

### 1.1. Challenges in Detecting Milk Adulteration

Because buffalo breeds are highly similar genetically, detecting foreign buffalo milk in commercial cheese is more complicated than identifying cow, goat, or sheep milk. Fraud involving cow milk is particularly concerning because it is cheaper and more widely available, especially during peak MdBC production seasons. These challenges highlight the need for highly specific and sensitive analytical methods to verify product authenticity.

### 1.2. Current Analytical Methods for Milk Authentication

Various analytical strategies have been developed to detect milk adulteration, differing in specificity and sensitivity. In Europe, the official regulatory method for detecting cow milk in buffalo cheese is isoelectric focusing (IEF) of γ-caseins after plasminolysis [1]. This method can detect cow milk adulteration at levels as low as 0.5% but does not distinguish between Mediterranean and foreign buffalo milk. Additionally, IEF does not differentiate sheep and goat γ-caseins, as their γ2- and γ3-casein sequences are identical. Recent advances in biomolecular techniques have shifted the focus toward proteomics-based methods for species identification. Mass spectrometry (MS), particularly MALDI-TOF-MS, provides high sensitivity and specificity for detecting cow, buffalo, goat, and sheep milk in complex dairy matrices [2]. Polymerase chain reaction (PCR) has also been proposed for detecting bovine DNA in buffalo mozzarella, with detection limits as low as 0.1–1% [3,4,5]. Additionally, genotyping techniques have been developed to differentiate imported buffalo milk from authentic Italian Mediterranean buffalo milk [6].

### 1.3. Casein Genetic Polymorphism and Geographic Origin Verification

Italian Mediterranean buffaloes have undergone centuries of selective breeding without crossbreeding, leading to distinct genetic traits that enable geographic origin verification. Comparative proteomics studies between Mediterranean and foreign buffalo breeds have revealed structural differences in β-casein (β-CN) and α_s1_-casein (α_s1_-CN).

β-CN variant B is predominant in Italian Mediterranean buffaloes, while foreign buffalo breeds also express β-CN A, distinguishable by specific amino acid substitutions (e.g., Thr41 (A) → Met41 (B) and Asn68 (A) → Lys68 (B)) [7]. Additional variations include Ser10 (A) → Gly10 (B) [8].A unique α_s1_-CN variant containing an internal deletion (residues 35–42) is absent in Italian Mediterranean buffaloes but present in foreign buffalo breeds [8,9,10].

The presence of the β-casein variant A or α_s1_-CN (35–42) deletion strongly suggests the use of non-PDO buffalo milk.

### 1.4. Objectives of the Study

This study aims to develop a robust protocol for determining the geographical origin of milk used in PDO cheese production and to detect the illicit addition of non-buffalo milk. The methodological approach integrates the following elements:Official European regulatory methods (IEF-based detection)Alternative analytical techniques, such as gel electrophoresis followed by immunodetection with polyclonal antibodies and mass spectrometry-based analyses (MALDI-TOF-MS and nano-LC-ESI-MS/MS).

By combining electrophoretic, immunological, and proteomics approaches, this study enhances the accuracy and sensitivity of milk authenticity verification, ensuring greater traceability and compliance with PDO standards.

## 2. Materials and Methods

### 2.1. Materials

Ultrapure water was obtained using a Milli-Q system (Millipore, Bedford, MA, USA). Macro-Prep Ceramic Hydroxyapatite Type I (HA) was sourced from Bio-Rad (Milan, Italy). Analytical grade tris (hydroxymethyl) aminomethane hydrochloride (Tris–HCl), urea, potassium chloride, trifluoroacetic acid (TFA), orthophosphoric acid (PA), acetonitrile (ACN) for HPLC Plus, and ammonium bicarbonate buffer (AMBIC) were obtained from Carlo Erba (Milan, Italy). Sequencing-grade modified trypsin was purchased from Promega (Madison, WI, USA). Analytical grade sinapinic acid, 2,5-dihydroxybenzoic acid (DHB), and sodium acetate trihydrate were supplied by Fluka (St. Louis, MO, USA). Analytical grade TEMED, ammonium persulphate. NaH_2_PO_4_, NiCl_2_, glycerol, trichloroacetic acid, methanol, acetic acid, Coomassie Brilliant Blue (G250) and equine serum were purchased from Sigma Aldrich (Darmstadt, Germany).

### 2.2. Sample Preparation

Sixty-four anonymized PDO MdBC and foreign mozzarella samples were provided by the Italian Central Inspectorate for Fraud Repression and Quality Protection of the Agrifood Products and Food, Ministry of Agricultural and Forestry Policies (Rome, Italy). PDO MdBC samples were labelled as IM and TU from 1 to 25. 

Three biological replicates of fresh milk samples from various origins were skimmed by centrifugation at 3220× *g* for 30 min, at 4 °C. Casein was isolated via isoelectric precipitation [2], separated from the whey, washed with distilled water, dissolved at pH 7.0 in 0.1 N NaOH, freeze dried, and stored at −20 °C until use.

Mozzarella cheese (10 g each) and water buffalo blue cheese (2 g each) samples were homogenized with 45 mL of 0.1 N HCl in a Stomacher bag using an Ultra Turrax T50 homogenizer (Janke & Kunkel IKA^®^ Labortechnik, Staufen, Germany) for 1 min. The resulting protein solution was defatted by centrifugation at 3220× *g* for 30 min, at 4 °C, adjusted to pH 4.6 with 1 M NaOH, and left to stand for 30 min. The precipitate was recovered by centrifugation, washed twice with distilled water, and washed with cold 1:4 acetone/water and diethyl ether to remove residual fat. The solvent-free dry precipitate was stored at −20 °C until analysis. Supernatants from the water buffalo blue cheese samples were recovered, freeze-dried, and stored at −20 °C for subsequent analysis.

### 2.3. Isoelectric Focusing and Immunoblotting of Caseins

Dry casein samples from fresh milk and mozzarella cheese were dissolved in a 9 M urea solution (10 g/L) containing 2-mercaptoethanol (1 mL/L). A 7 μL aliquot was applied to ultra-thin layer isoelectric focusing (UTLIEF; gel thickness 0.25 mm) with a pH gradient of 2.5–6.5, slightly modified with respect to the European reference method [1]. Ampholine (pH 2.5–5, 4.5–5.4, and 4–6.5) (GE Healthcare, Hatfield, UK) was mixed in a 1.6:1.4:1 (*v*/*v*/*v*) ratio. Focusing was performed using a 2117 Multiphor II Apparatus (LKB, Bromma, Sweeden) at 10 °C with the following conditions: pre-focusing: 2000 V, 15 mA, 4 W, 30 min; sample focusing: 2000 V, 15 mA, 4 W, 60 min; final focusing: 3000 V, 5 mA, 20 W. Protein bands were stained with Coomassie Brilliant Blue (G250) [11]. After UTLIEF, proteins were transferred onto nitrocellulose membranes (Trans-Blot, 0.45 μm, Bio-Rad) by capillary diffusion. Immunoblotting was performed using rabbit polyclonal antipeptide antibodies raised against bovine α_s1_-CN (1–22), β-CN (195–209) and β-CN (106–110) (Proteogenix, Schiltigheim, France) [8,12]. Theoretical isoelectric point (pI) values were calculated using the pI/Mw tool available at the Expasy site (web.expasy.org/compute_pi (accessed on 7 February 2024).

### 2.4. Trypsin Hydrolysis

Isoelectric casein and the insoluble fraction of buffalo cheese samples (1 mg each) were dissolved in 100 μL of 0.4% (*w*/*v*) AMBIC buffer (pH 8.0) and incubated with sequencing-grade trypsin at a 50:1 (casein: enzyme) for 4 h, at 37 °C. A 20 μL aliquot of the digest was desalted using ZipTipC18^®^ C18 (Millipore, Bedford, MA, USA) before MALDI-TOF-MS and nano-LC-ESI-MS/MS analysis.

### 2.5. HA-Based Enrichment of Casein Phosphopeptides

Isoelectric casein (1 mg) was dissolved in 1 mL of 50 mM Tris–HCl (pH 7.8) containing 4.5 M urea, 0.2 M KCl, and 10 mM DTT, then digested in situ with trypsin as described above. The digest was loaded onto HA microspheres (10 mg) pre-equilibrated in the same buffer. After 30 min of gentle shaking at room temperature, unbound peptides were removed by successive washes with (i) the loading buffer, (ii) 50 mM Tris–HCl (pH 7.8), (iii) 20 mM Tris–HCl (pH 7.8), containing 20% ACN, and (iv) Milli-Q water. The HA microspheres were dried using a SpeedVac device (Thermo Electron, Milford, MA, USA). Bound casein phosphopeptides (CPPs) were recovered by dissolving HA particles in 5% (*w*/*v*) phosphoric acid, followed by ZipTip^®^ C18 clean-up before MS analysis [13].

### 2.6. Limit of Detection for Foreign Buffalo Casein

Calibration curves were prepared using triplicate mixtures of Mediterranean buffalo milk spiked with Venezuelan buffalo milk at concentrations of 50%, 25%, 12.5%, 6.25%, 3.125%, 1.56%, and 0.78% *v*/*v*. Isoelectric casein from each mixture was trypsinolyzed, enriched for CPPs on HA, and analyzed by MALDI-TOF-MS and nano-LC-ESI-MS/MS to detect buffalo β-CN A and B peptides, as well as the internally deleted α_s1_-CN (43-58). Two signature peptides, α_s1_-CN (43-58)2P (MH^+^ signal at *m/z* 1913.5) and β-CN (1-28)4P (MH^+^ signal at *m/z* 3488.4), defined the minimum detection threshold. Although signals were observed below 0.78% *v*/*v* contamination, this concentration was established as the practical limit of detection for foreign milk adulteration in PDO MdBC cheese.

### 2.7. Limit of Detection for Bovine Casein

A similar calibration curve was generated using a triplicate mixture of Mediterranean buffalo milk spiked with bovine milk at concentrations of 50%, 25%, 12.5%, 6.25%, 3.125%, 1.56%, and 0.78% *v*/*v*. The isoelectric casein was digested, CPPs were enriched on HA, and the samples were analyzed by MALDI-TOF-MS and nano-LC-ESI-MS/MS to detect the bovine-specific peptides α_s1_-CN (8-22) (MH^+^ signal at *m*/*z* 1759.9) and β-CN (1-25)4P (MH^+^ signal at *m*/*z* 3122.2). The highest dilution at which these peptides could not be detected was considered the limit of detection.

### 2.8. Casein Plasminolysis

The pH 4.6 insoluble fraction of defatted buffalo blue cheese samples was dissolved in 0.2 M ammonium bicarbonate (pH 8.0) and hydrolyzed with plasmin (Boehringer, Mannheim, Germany) at 37 °C, using an enzyme-to-substrate (E:S) ratio of 1:1000. After 1 h, the reaction was stopped by adding trichloroacetic acid to a final concentration of 12% *w/v*. The mixture was centrifuged at 12,000 rpm for 5 min, and the resulting precipitate was washed thrice with distilled water. This material was dissolved in 6 M urea and analyzed by UTLIEF following the above procedure.

### 2.9. MALDI-TOF-MS Analysis of Casein Peptides

Tryptic digests and isolated CPPs were analyzed using an UltrafleXtreme MALDI-TOF-MS (Bruker Daltonics, Bremen, Germany). Samples were mixed 1:1 (*v*/*v*) with either 10 mg/mL DHB (Bruker Daltonics, Bremen, Germany) in 30% *v/v* ACN/0.1% *v/v* TFA for non-phosphorylated peptides or 10 mg/mL DHB in ACN/water/phosphoric acid (50:49:1, *v*/*v*/*v*) for CPPs. The mixtures were then spotted onto a ground steel plate and air-dried. Spectra were acquired in positive-ion reflectron mode (laser frequency, 1000 Hz; ion source 1 voltage, 25.19 kV; ion source 2 voltage, 23.94 kV; lens voltage, 6.50 kV; sample rate, 0.16; *m*/*z* 500–5000). External mass calibration was performed using Peptide Calibration Standard II (Bruker Daltonics, Bremen, Germany). Each sample was run at least in triplicate, and spectra were processed with FlexControl 3.4 software package (Bruker Daltonics), which included spectral mass adjustment (compression by a factor of 10 in the total mass range), optional smoothing (using the Savitsky−Golay algorithm with a frame size of 25 Da), spectral baseline subtraction, normalization, internal peak alignment, and peak picking. Pretreated data were then subjected to visualization and statistical analysis. In all cases, spectral acquisition methods were developed to maximize the number of signals present in the mass spectra and the corresponding signal-to-noise ratios. MALDI-TOF mass signals were assigned to specific components using GPMAW 4.23 software (Lighthouse Data, Denmark), which generated a mass/fragment database based on protein sequence and protease specificity. All peptide assignments were confirmed manually by inspecting the corresponding mass spectra. Peptides showing a statistically significant difference in signal intensity or mass value were determined.

### 2.10. Recovery of the Peptide Fraction from the Water-Soluble Extract of Buffalo Blue Cheese

The pH-soluble fraction from buffalo blue cheese samples was reconstituted in 500 µL of MilliQ water and fractionated using 10 kDa cut-off ultrafiltration cartridges (Amicon^®^, Ultra Centrifugal Filter, Sigma Aldrich, Darmstadt, Germany). The 10 kDa permeate fraction was freeze-dried, redissolved in a water solution containing 0.1% *v/v* TFA and desalted using a C18 ZipTip^®^ microcolumns (Millipore, Bedford, MA, USA). Peptide quantification followed the protocol by Arena and coworkers [14], using the Quantitative Colorimetric Peptide Assay™ (Thermo Scientific Pierce, Waltham, MA, USA).

### 2.11. Nano-LC-ESI-MS/MS Analysis of Peptide Mixtures

Nano-LC-ESI-MS/MS was performed on 100 ng of peptide mixtures using an Ultimate 3000 ultra-high-performance nano-liquid chromatography system (ThermoFisher Scientific, San Jose, CA, USA) coupled to a Q-Exactive Orbitrap Plus mass spectrometer (ThermoFisher Scientific, Bremen, Germany). Peptides were injected via the auto-sampler onto an EASY-Spray™ PepMap RP C18 column (15 cm × 75 µm, 3 µm particle diameter, 100 Å pore size) (ThermoFisher Scientific) and eluted at 300 nL/min with 0.1% *v*/*v* formic acid (solvent A) and 0.1% *v*/*v* formic acid in acetonitrile (solvent B). The column was equilibrated at 3% solvent B and subjected to a linear gradient from 3 to 45% B over 60 min. The mass spectrometer operated in data-dependent acquisition mode, with MS scans (*m*/*z* 350–1600) acquired in positive ion mode. The top 10 ions in MS were selected for MS/MS fragmentation. Precursor spectra were generated at a resolving power of 70,000 full width at half maximum (FWHM), with automatic gain control (AGC) targets of 1 × 10^6^ and 1 × 10^5^ ions for full MS and MS/MS spectra, respectively, and a maximum ion injection time of 100 ms. Mass fragmentation spectra were obtained at a resolving power of 17,500 FWHM, with a dynamic exclusion of 10 s. Parent ions with a net charge greater than 6 were excluded from the selection for fragmentation.

Data analysis was performed using Xcalibur software v. 3.1 (ThermoFisher Scientific). Raw nano-LC-ESI-MS/MS files were uploaded into the Andromeda search engine in the MaxQuant bioinformatics suite (v. 1.6.2.10) [15]. Biological replicates of each MdBC or buffalo blue cheese sample were indexed within Andromeda for proper handling [15]. Peptide quantification used Andromeda’s label-free quantification (LFQ) option. Two distinct database searches were performed against *Bubalus bubalis* and *Bos taurus* sequences from the UniProtKB database, supplemented by a manually curated *B. taurus* database [16] for enhanced specificity. The following parameters were applied to all raw data searches: a mass tolerance value of 10 ppm for the precursor and 0.02 for the fragment ions; no proteolytic cleavage specificity; variable modifications including Met oxidation, pyroglutamic acid formation at N-terminal Gln, and Ser/Thr phosphorylation. Peptide spectrum matches (PSMs) were filtered using a target decoy database approach with an e-value of 0.01 and peptide-level false discovery rate (FDR) of 1% (e-value of 0.01), corresponding to a 99% confidence score. For final data interpretation, the peptide.txt output was filtered to retain only peptides confidently identified in all three biological replicates of the same cheese sample at a given time, along with their corresponding PSMs. Measured ion intensities (ion counts) were averaged across replicates. Intra-replicate reproducibility was assessed at both peptide and protein levels via Pearson correlation using Perseus (v. 1.6.15.0) [17].

### 2.12. Statistical Analysis

MdBC cheese samples were analyzed in technical triplicate, obtaining three independent measurements. Data are presented as mean ± standard deviation. Statistical evaluations were carried out GraphPad Prism software version 6.0 (GraphPad Software Inc., San Diego, CA, USA). The proteotypic signals distinguishing buffalo, bovine, and ovine milk proteins were assessed based on MS ion intensity ratios (bovine/buffalo and ovine/buffalo). Calibration curves were constructed utilizing raw MS intensity data of selected proteotypic peptides to establish contamination levels. Among the evaluated proteotypic peptides, particular emphasis was placed on peptide α_s1_-CN (8-22) from both species (bovine and buffalo MH^+^ signals at *m*/*z* 1687.9 and *m*/*z* 1759.9, respectively). Differences in peptide intensities among groups were statistically evaluated through one-way ANOVA followed by Tukey’s HSD test, considering *p*-values <0.05 statistically significant.

## 3. Results

### 3.1. Gel Isoelectric Focusing of Buffalo Casein for Milk Origin Identification

We used gel-based isoelectric focusing to detect possible adulteration of PDO MdBC cheese with foreign buffalo milk. As shown in Figure 1A (Coomassie R-250 staining), distinct casein profiles were observed for both Mediterranean and foreign buffalo milk as well as the corresponding mozzarella cheeses. Isoelectric focusing allowed clear differentiation of β-CN variant A, which can harbor five or six phosphate groups, producing bands at characteristic isoelectric point (pI) values. These bands were readily distinguishable from that of β-CN variant B from animals having Mediterranean origin, enabling the identification of buffalo milk from different geographical sources. However, distinguishing the Mediterranean α_s1_-CN variant and the foreign internally deleted α_s1_-CN (35–42) variant was more challenging because both carry seven phosphate groups. Immunoblotting with polyclonal antipeptide antibodies enhanced the detection of α_s1_-CN and β-CN bands, increasing the selectivity and sensitivity compared with non-specific Coomassie staining. By targeting bovine α_s1_-CN (1-22) and β-CN (195-209), which share identical primary structure in Mediterranean and foreign buffalo, these antibodies reliably detected parent caseins from both sources (Figure 1B,C). In buffalo–bovine milk mixtures, distinguishing bovine β-CN variants from buffalo β-CN variant B 5P remained difficult because both have nearly identical theoretical pI values (~4.66). Consequently, gel-based isoelectric focusing is unsuitable for detecting bovine β-CN in these conditions. In contrast, foreign buffalo β-CN variant A features three amino acid substitutions (e.g., Ser10 (A) → Gly10 (B), Thr41 (A) → Met41 (B) and Asn68 (A) → Lys68 (B) and exhibits two phosphorylation states (five and six phosphate groups). These differences generate theoretical pI values of 4.50 and 4.57, respectively, which differ significantly from β-CN variant B 5P (pI = 4.66). Accordingly, the exclusive presence of β-CN A 5P and β-CN A 6P in samples 3, 4, and 6 (Figure 1) underscores their utility as molecular markers for identifying foreign buffalo breeds and tracing the geographic origin of milk used in MdBC cheese production. Polyclonal antibodies against bovine β-CN (195-209) further confirmed the separation of β-CN A 5P and 6P (Figure 1C), with no evidence of co-migrating proteins. These findings demonstrate that gel-based isoelectric focusing combined with conventional staining provides a cost-effective and relatively straightforward method for detecting β-CN A phosphoforms (containing five and six phosphate groups) in routine quality control. Despite these advantages, the method is less effective at identifying the foreign α_s1_-CN 7P variant with the internal deletion (35–42). Theoretical calculations indicate that its 7P forms (pI~4.32 and~4.33) migrate closely on the gel (Figure 1A). Even when immunoblotting utilized polyclonal antisera against bovine α_s1_-CN (1-22), distinguishing the internally deleted α_s1_-CN—a critical genetic marker of foreign buffalo milk—remained difficult under standard forensic isoelectric focusing conditions without specialized peptide-directed antibodies.

### 3.2. Gel Isoelectric Focusing of MdBC Cheese Samples for Detection of Food Adulteration

An example of UTLIEF analysis of mozzarella cheese samples stained with Coomassie Brilliant Blue G250 is shown in Figure 2A. Compared with foreign buffalo casein (lane 3), PDO MdBC samples (lane 4-17) generally showed no evidence of adulteration due to the addition of non-Mediterranean buffalo milk. Key findings from the UTLIEF profiles include that sample IM22 (lane 4) displayed significant contamination with bovine casein, indicated by the presence of β-CN A1. This result was confirmed by comparison with a known bovine β-CN heterozygous standard (A1/A2) (lane 1). Densitometric analysis of α_s1_-CN bands revealed that bovine α_s1_-CN constituted approximately 37% of the total α_s1_-CN fraction. Across all samples analyzed, whether stained with Coomassie Brilliant Blue or immunostained with the polyclonal antipeptide antibody targeting β-CN (195-209) (Figure 2C), no β-CN from foreign buffalo milk was detected. 

Mass spectrometry analysis was conducted to confirm and quantitatively assess the presence of foreign caseins in samples suspected of adulteration. To validate these findings, a set of proteotypic peptides, which serve as surrogates for their parent caseins, was initially screened by MALDI-TOF-MS and subsequently confirmed via nano-LC-ESI-MS/MS (see below). This bottom-up proteomic approach involved tryptic digestion of recovered casein from MdBC cheese samples, followed by MS analysis of the resulting peptides. The method provided extensive sequence information, enabling unambiguous peptide identification. Additionally, a curated subset of signature peptides, selected from the primary sequence of 16 casein fractions representing three dairy species (documented in the Peptide Atlas database), was used for accurate species identification.

### 3.3. MALDI-TOF-MS Analysis of MdBC Casein Digests for the Detection of Extraneous Milk

#### 3.3.1. Selection of Proteotypic Peptides for Casein Identification

Proteotypic peptides were selected for their uniqueness within the major ruminant caseome (cow, sheep, and buffalo), enabling them to serve as exclusive analytical surrogates for their parent caseins. However, the extensive sequence homology among caseins from Italian Mediterranean and foreign buffalo, as well as goat and sheep caseins, posed significant analytical challenges. In silico digestion of casein sequences, accounting for the predicted trypsin cleavage sites, identified three proteotypic peptides specific for β-CN variants and one peptide targeting the internally deleted (35–42) α_s1_-CN variant (Table 1). 

The selected peptides met the following criteria:A minimum length of 10 amino acids;A high stability, avoiding residues prone to artifactual modifications (e.g., methionine, cysteine, aspartic acid-glycine pairs, N-terminal glutamine, and asparagine).

Although Lys-Lys sites in α_s1_-CN and α_s2_-CN may result in incomplete trypsin digestion, these segments lie outside the regions responsible for producing proteotypic peptides and thus have minimal impact. Table 1 summarizes the theoretical identity, the phosphorylation level, the amino acid substitutions, and the calculated mass value of these peptides. The experimental validation of the in silico predictions included comparing the predicted peptides with those observed in the experiments, as deriving from tryptic digestion of caseins from various species.

#### 3.3.2. Evaluation of Bovine Casein in MdBC Cheese by MALDI-TOF-MS

A calibration curve was established using buffalo milk samples spiked with increasing proportions of cow milk (0.78–50%, *v*/*v*). Each casein fraction obtained at the isoelectric point was processed according to the protocol reported in the experimental section, and the resulting tryptic digests were analyzed via MALDI-TOF-MS. As depicted in Figure 3, the calibration curve relied on the relative signal intensity of proteotypic peptides α_s1_-CN (8–22) from both animal species, having MH^+^ signals at *m*/*z* 1687.9 and 1759.9 for the buffalo and bovine peptide, respectively; these peptides are distinguished by their molecular mass difference arising from Glu14 → Gly14 substitution. 

Observed mass difference (Δm = 72 Da) enabled a clear discrimination of the buffalo peptide versus the bovine counterpart by MALDI-TOF-MS. The calibration curve demonstrated a robust linear correlation within the tested range (0.78–50%), achieving a Pearson correlation coefficient greater than 0.98. Although further optimization could improve sensitivity, the quantification limit of 0.78% was considered adequate for verifying cheese authenticity. The measured sensitivity was comparable to that of the EU’s official regulatory method based on isoelectric focusing of γ-caseins, which identifies cow milk contamination at approximately 0.5% in buffalo mozzarella. Nonetheless, MALDI-TOF-MS-based method offers superior reliability, as the observed mass values closely match their theoretical values, ensuring precise identification of contaminant proteins. 

Additional calibration curves were tentatively developed by plotting peak intensity ratios of proteotypic peptides β-CN (33–48)P from bovine (MH^+^ signal at *m*/*z* 2061.8) versus buffalo (MH^+^ signal at *m*/*z* 2091.8), differing for Met41 → Thr41 substitution, and of bovine peptide β-CN (1–25)4P (MH^+^ signal at *m*/*z* 3122.3) versus buffalo β-CN (1–28)4P (MH^+^ signal at *m*/*z* 3458.4), the latter ones distinguished by Arg25 → His25 substitution. However, the coexistence within the same spectrum of phosphorylated/non-phosphorylated species or oxidized Met41 in buffalo peptides strongly limited the suitability of these calibration curves for accurate adulteration quantification. Nevertheless, the detection of above-reported peptides provided unequivocal evidence of bovine contamination.

#### 3.3.3. MALDI-TOF-MS Quantification of Adulteration in PDO MdBC Cheese

MALDI-TOF-MS analysis of the IM-labeled MdBC samples revealed the presence of bovine proteotypic peptides in twelve instances (IM1, IM7, IM9, IM11, IM13, IM14, IM15, IM16, IM17, IM19, IM22, and IM25), as summarized in Appendix A. Rather than preparing individual calibration curves or using synthetic peptides for absolute quantification, we adopted a simplified approach. Specifically, we calculated signal intensity ratios of the predicted proteotypic peptides, assuming that ion signal intensity is proportional to the absolute amount of each parent protein. This assumption relies on the premise that amino acid substitutions without net charge change generally do not significantly affect peptide ionization efficiency. Consequently, the relative intensities of orthologous peptides from different species in mixed samples reflected the absolute quantity of their parent proteins. Through Flex Analysis software (Bruker Daltonics, Bremen, Germany), we compared peptide ion intensities of proteotypic peptides including β-CN (33–48)P (bovine MH^+^ signal at *m*/*z* 2061.8 versus buffalo MH^+^ signal at *m*/*z* 2091.8) (Met41 → Thr41 substitution), and bovine β-CN (1–25)4P (MH^+^ signal at *m*/*z* 3122.3) versus buffalo β-CN (1–28)4P (MH^+^ signal at *m*/*z* 3458.4) (Arg25 → His25 substitution). Both proteotypic peptides were detected in adulterated buffalo mozzarella sample IM22, whereas only bovine β-CN (33–48)P (MH^+^ signal at *m*/*z* at 2061.8) was identified in samples IM14, IM16, and IM19. Surprisingly, the MALDI-TOF-MS spectrum of the tryptic digest of casein from sample IM22 revealed that the signal intensity of the bovine peptide β-CN (1–25)4P corresponded to 60.8% adulteration. On the other hand, MALDI-TOF-MS analysis of six samples of MdBC cheeses showed bovine peptide α_s1_-CN (8–22) percentages ranging from 0.1 to 0.8%, except for IM22, which coherently exhibited a significantly higher level. Detection of additional proteotypic peptides, including bovine α_s1_-CN (133–151) (MH^+^ signal at *m*/*z* 2316.2) and β-CN (49–68) (MH^+^ signal at *m*/*z* 2223.2), further indicated bovine adulteration of MdBC cheeses. Surprisingly, samples IM9 and IM11 also contained ovine-specific peptides α_s1_-CN (23–33) (MH^+^ signal at *m*/*z* 1307.7) and α_s1_-CN (133–151) (MH^+^ signal at *m*/*z* 2328.2), with the last indicating approximately 7% adulteration with sheep’s milk. Conversely, sample IM25 contained only ovine peptide α_s1_-CN (23–33).

Appendix A summarizes the proteotypic peptides observed in MdBC cheeses of the TU series. TU17 was the only sample showing evidence of milk adulteration. It contained only ovine proteotypic species, specifically peptides α_s1_-CN (8–22) (MH^+^ signal at *m*/*z* 1718.9), α_s1_-CN (23–33) (MH^+^ signal at *m*/*z* 1307.7), and α_s1_-CN (133–151) (MH^+^ signal at *m*/*z* 2328.2), with the first indicating an adulteration level of approximately 0.2%.

### 3.4. Validation of Adulteration of MdBC Cheese by Independent MS Methods

To achieve the highest confidence and reliability in quantifying extraneous milk in buffalo cheese, we conducted complementary experiments based on nano-LC-ESI-MS/MS analysis of tryptic digests of casein from MdBC cheese samples as well as MALDI-TOF-MS-based protein fingerprinting of the corresponding phosphopeptides.

#### 3.4.1. Proteomic Identification of MdBC Cheese Adulterations by Nano-LC-ESI-MS/MS

To validate the detection and quantitation of bovine casein in MdBC cheese samples, we analyzed their tryptic digests by nano-LC-ESI-MS/MS. Among the IM samples previously classified as free of extraneous buffalo proteins by UTLIEF, twelve contained various bovine proteotypic peptides (IM1, IM7, IM9, IM11, IM13, IM14, IM15, IM16, IM17, IM19, IM22, and IM25), confirming MALDI-TOF-MS results. These peptides are indicated with the symbol “●” in Appendix A; additional results are shown in Appendix A. This nano-LC-ESI-MS/MS approach not only confirmed the presence of the bovine peptide α_s1_-CN (8-22)—previously detected by MALDI-TOF-MS in several MdBC cheese samples (IM9, IM11, IM13, IM15, IM22, and IM25)—but also demonstrated superior sensitivity and accuracy in detecting and quantifying this molecular target. In addition, novel bovine α_s1_-CN-derived peptides, including α_s1_-CN (37-58)2P (MH^+^ signal at *m*/*z* 2598.1), were identified in samples IM1, IM8, IM12, IM14, IM16, and IM17, despite they escaped detection by other analytical techniques. 

On the other hand, MdBC samples TU6, TU14, TU15, TU17, TU18, TU20, TU22, TU23, and TU25 were ascertained to contain adulterating bovine milk through the detection of specific proteotypic peptides (see Appendix A). In addition to bovine adulteration, partial substitution of buffalo milk with the ovine equivalent was confirmed to occur in sample TU17, as evidenced by the detection of various sheep-specific proteotypic peptides, including α_s1_-CN (133–151) (MH^+^ signal at *m*/*z* 2328.2), κ-CN (35–68) (MH^+^ signal at *m*/*z* 4065.1), κ-CN (69–86) (MH^+^ signal at *m*/*z* 1947.1), and β-CN (133–169) (MH^+^ signal at *m*/*z* 4165.3). Also sample TU20 showed evidence of ovine adulteration, as indicated by β-CN (114-132) (MH^+^ signal at *m*/*z* 2183.2). Altogether, these results suggested the occurrence of multiple sources of adulteration in various MdBC cheeses.

#### 3.4.2. Evaluation of Extraneous Casein in MdBC Cheese by Phosphoproteomics

Phosphoproteomics offers a promising approach for authenticating MdBC cheeses by identifying species- and buffalo breed-specific markers. This advanced technique can enable the detection of adulteration by focusing on proteotypic phosphopeptides generated following trypsinolysis of α_s1_-CN and β-CN (Table 1). These markers allow differentiation among caseins from various animal species and buffalo breeds, providing insights onto the adulteration of PDO MdBC cheese. However, MALDI-TOF-MS-based phosphopeptide analysis without selective molecular enrichment suffers from weak signals due to low abundance of phosphorylated components species, and peak suppression by coexisting non-phosphorylated species. Initial MALDI-TOF-MS analyses of casein tryptic digests of MdBC samples without phosphopeptide enrichment revealed very few CPP signals, hindering species or breed differentiation (Figure 4). To address this limitation, we performed hydroxyapatite (HA)-based absorption chromatography to effectively isolate CPPs from the above-reported digests by exploiting phosphate group interactions with immobilized Ca^2+^ ions, while subsequent washing with a 0.01 M phosphate buffer minimized nonspecific binding and enhanced analytical precision. This allowed detecting adulteration due to illegal addition of bovine milk in PDO MdBC cheeses. The bovine peptide α_s1_-CN (8-22)P was decisive, appearing in 7 out of 50 MdBC cheese samples by both MALDI-TOF-MS and nano-LC-ESI-MS/MS, and in 9 additional samples solely by nano-LC-ESI-MS/MS, thereby confirming the superior sensitivity of the latter mass spectrometric technique. Detection of additional bovine proteotypic peptides, such α_s1_-CN (133–151) (MH^+^ signal at *m*/*z* 2316.1), α_s1_-CN (106-119)P (MH^+^ signal at *m*/*z* 1660.8), and α_s1_-CN (104-119)P (MH^+^ signal at *m*/*z* 1951.9), further validated adulteration in multiple MdBC samples (IM1, IM7, IM9, IM11, IM13, IM14, IM15, IM16, IM17, IM19, IM22, IM25, and TU18).

Casein tryptic digests of MdBC cheese samples selectively enriched for CPPs and analyzed by MALDI-TOF-MS were further investigated for the occurrence of buffalo breed-specific phosphopeptide markers of the addition of foreign buffalo milk. For this purpose, α_s1_-CN and β-CN phosphopeptides α_s1_-CN (23–34) Met31 (MH^+^ signal at *m*/*z* 1416.7), α_s1_-CN (35–58)P with the internal deletion (35–42) (MH^+^ signal at *m*/*z* 1913.7), β-CN (33–48)P (MH^+^ signal at *m*/*z* 2061.8), β-CN (1–28)4P (MH^+^ signal at *m*/*z* 3488.5), and β-CN (1–28)5P (MH^+^ signal at *m*/*z* 3568.5) (Table 1) were searched in the corresponding mass spectra. Among all samples investigated, only cheese IM7 and a foreign curd reference sample contained α_s1_-CN (23–34) Met31 and α_s1_-CN (35–58)P with the internal deletion (35–42), confirming UTLIEF results (see above) on the general absence of non-Mediterranean buffalo milk in the MdBC samples assayed in this study.

### 3.5. UTLIEF Coupled to Immunodetection May Fail in Revealing Extraneous Milk in Buffalo Blue Cheese

UTLIEF analysis combined with immunodetection with polyclonal antipeptide antibodies significantly enhances the detection sensitivity and the limit of recognition of extraneous milk in buffalo cheeses [18], compared to the official European regulatory method. As shown in Figure 5, this approach leverages the distinct isoelectric points of γ2-caseins from cow and buffalo milk, which arise from the specific amino acid substitution (Pro148 → His148 in buffalo vs. bovine β-CN) [19].

The analysis involves visual inspection or densitometric quantification of the bovine γ2-casein band intensity in test samples, relative to cheese samples with known cow’s milk content. The results shown in Figure 5 indicate that commercial MdBC samples adulterated with cow’s milk exhibited bovine γ2-casein intensities exceeding the 0.5% threshold revealed with reference bovine-buffalo milk mixtures, definitively assigning their non-compliance. In contrast, a co-analyzed sample of buffalo blue cheese subjected to 65 days of maturation apparently showed no detectable bovine γ2-casein. Alternative buffalo milk-based products like buffalo blue cheeses might represent a challenge for the European official regulatory method because extensive fungal-driven proteolysis degrades γ-caseins, which serve as species-specific markers. As a result, the official method might become unsuitable for detecting bovine adulteration in these cheese typologies. To further investigate this issue, a complementary analytical approach using nano-LC-ESI-MS/MS was developed and applied to the above-reported buffalo blue cheese sample with the aim of evaluating the certainty of the corresponding UTLIEF-immunodetection results.

### 3.6. Identification of Bovine Material in Buffalo Blue Cheese by Nano-LC-ESI-MS/MS

The inherent complexity of buffalo blue cheese, shaped by the extensive proteolysis due to *Penicillium roqueforti* action, poses unique analytical challenges to ascertain food adulterations. During ripening, fungal enzymes significantly proteolyze casein, yielding a peptide-rich soluble fraction. To investigate buffalo blue cheese adulterations, a comprehensive peptidomic-proteomic strategy was developed, which is based on nano-LC-ESI-MS/MS analysis of (i) the <10 kDa permeate of the cheese water extract at pH 4.6, and (ii) the tryptic digest of the corresponding pH 4.6 insoluble fraction. Appendix A provide a detailed list of the peptides assigned in the former and the latter case, respectively, and the corresponding parent proteins. These analyses confirmed the presence of bovine milk-derived peptides in the soluble and insoluble fractions of the above-reported buffalo blue cheese, which otherwise provided negative findings following combined UTLIEF-immunodetection analysis. Comprehensive results are summarized in Table 2. 

The water-soluble fraction contained more bovine peptides (65 in number) than the water-insoluble fraction (11 in number), illustrating the advantage of peptidomics in dissecting highly proteolyzed cheese matrices, with respect to UTLIEF combined with immunodetection. Notably, 14, 29, and 14 peptides derived from bovine α_s1_-CN, α_s2_-CN, and β-CN A1 variant, respectively, were detected in the soluble fraction, along with 7 peptides from the β-CN A2 variant. Overall, peptidomics proved more effective than conventional proteomics in revealing species-specific differences in cheeses maturated under conditions of extensive proteolysis.

### 3.7. Peptidomic Quantitation of Bovine Material in Buffalo Blue Cheese

Within the pH 4.6 water-soluble extract of buffalo blue cheese, numerous β-CN-derived peptides were identified (Appendix A); we detected coexisting orthologous peptides from bovine and buffalo sources. Marker peptides from the A1 variant of bovine β-CN (His67-Asn68 motif) were distinguished from those of the B variant of buffalo β-CN (Pro67-Lys68 motif). By examining multiple pairs of orthologous β-CN-derived peptides, the relative quantities of bovine β-CN in the buffalo counterpart were estimated by nano-LC-ESI-MS/MS analysis through a MaxQuant software-assisted comparison of the corresponding mass signal intensities (Table 3). This intensity ratio evaluation indicated that bovine β-CN A1 in the buffalo blue cheese sample ranged from 0.01% to 0.15%, with an overall mean of 0.04%. Considering the average protein content of cow’s milk (3.25%) and buffalo’s milk (4.70%)—a 1.45-fold difference—this ratio corresponded to approximately 3% (*v*/*v*) cow’s milk adulteration in buffalo blue cheese. Direct comparisons based exclusively on orthologous peptide intensities between bovine and buffalo β-CN may not fully represent actual species proportions, particularly when γ-caseins or other insoluble fractions are not considered. Furthermore, calibrations with synthetic orthologous peptides might not fully account for the inherent variability of bovine β-CN A1/A2 ratios or differences in proteolytic susceptibility between bovine and buffalo β-CN variants. Nevertheless, this peptidomic approach, focusing on multiple well-chosen markers rather than a single peptide, can provide significantly improved accuracy of adulteration quantification in cheeses maturated under conditions of extensive proteolysis, when compared to conventional electrophoretic and MALDI-TOF-MS-based methods.

### 3.8. Considerations for Real-Time Detection at Production Sites

While the analytical techniques used in this study—gel electrophoresis, MALDI-TOF-MS, and nano-LC-ESI-MS/MS—offer high specificity and sensitivity for detecting extraneous milk proteins, they are not designed for rapid, real-time screening at dairy production sites. In fact, these methodologies require specialized laboratory equipment, trained personnel, and extended analysis times, making them more suitable for confirmatory testing in centralized facilities.

To reduce the risk of accepting adulterated buffalo milk, PDO MdBC cheese producers frequently use immunochromatographic strip tests that detect bovine immunoglobulins (IgGs) in raw buffalo milk. This simple and rapid test provides results within 10 min and can detect bovine milk contamination at levels as low as 1%. However, this method detects only cow milk adulteration and cannot identify milk from other dairy species, such as sheep, goat, or foreign buffalo breeds.

Given these constraints, a multi-tiered control strategy is necessary. Rapid on-site screening using immunochromatographic tests can function as a first-line defense against fraudulent practices, enabling dairy processors to reject contaminated milk upon receipt. For comprehensive verification—particularly in cases where non-bovine adulteration is suspected—electrophoretic techniques and mass spectrometry-based proteomics remain essential for official regulatory testing and PDO cheese certification enforcement.

### 3.9. Comparative Analysis of Adulteration Detection Methods

Table 4 provides a comparative overview of the analytical techniques for detecting non-buffalo milk in MdBC cheese. Each method has distinct advantages and limitations concerning specificity, sensitivity, and applicability. While gel electrophoresis with polyclonal antibodies allows for rapid initial screening, mass spectrometry-based approaches, such as MALDI-TOF-MS and nano-LC-ESI-MS/MS, offer greater specificity and quantitative accuracy. Immunochromatographic test strips and HPLC detection of β-lactoglobulin A act as practical on-site screening tools for bovine whey proteins, though they are limited to detecting bovine adulteration. Some dairies employ the official Italian HPLC method to identify bovine β-lactoglobulin A [20], ensuring the authenticity of raw buffalo milk supplied in bulk tanks. This rapid assay provides reliable data on contaminating bovine whey proteins, essential for determining whether a milk supply is accepted or rejected. Integrating multiple methodologies strengthens fraud detection strategies, enhancing traceability and compliance with PDO cheese regulations.

## 4. Conclusions

This study demonstrates the effectiveness of multiple analytical strategies—including electrophoretic, immunochemical, spectrometric, and proteomic techniques—in detecting milk adulteration in PDO buffalo cheeses. Each method offers distinct advantages and limitations, emphasizing the need for a multifaceted approach to ensure the accurate identification and quantification of adulterants. Electrophoretic techniques, such as UTLIEF, are particularly useful for peripheral laboratories close to production sites, providing rapid preliminary screening of PDO buffalo cheese authenticity. MALDI-TOF-MS enhances this process by enabling the detection of proteotypic peptides from multiple dairy milk species within a single spectrum. However, nano-LC-ESI-MS/MS remains the gold standard for definitive fraud quantification, offering unparalleled specificity, sensitivity, and reliability. This technique is particularly advantageous for confirming milk adulteration in buffalo cheeses subjected to extensive proteolysis, where fungal-driven degradation complicates protein and peptide identification.

Looking ahead, advances in targeted proteomics, particularly predefined proteotypic peptide multiple reaction monitoring (MRM), will enable a more detailed profiling of casein markers. This will support a highly sensitive and specific approach for identifying undeclared milk ingredients in PDO buffalo cheeses. Moreover, ongoing progress in peptidomics continues to expand our ability to detect species- and breed-specific markers, even under the challenging conditions imposed by fungal-driven proteolysis. These analytical advancements not only reinforce consumer confidence but also safeguard the authenticity of PDO-certified products in the global market. Continued technological and methodological innovations will further improve fraud detection, guaranteeing improved food quality and safety standards worldwide.

## Figures and Tables

**Figure 1 foods-14-01193-f001:**
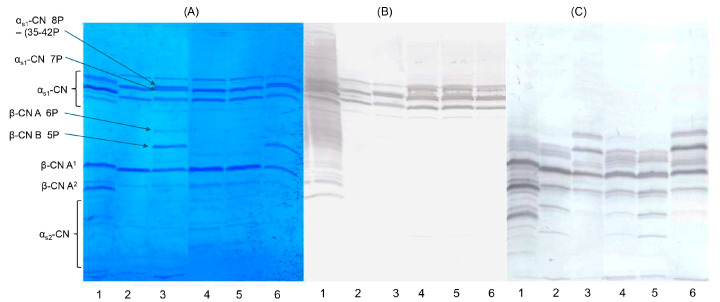
UTLIEF on a polyacrylamide gel (pH gradient 2.5–6.5) of casein samples from different sources. Lane assignment: (1) cow milk (β-CN A1/A2); (2) Italian Mediterranean buffalo milk; (3) Romanian buffalo milk; (4) Romanian mozzarella cheese; (5) Mediterranean PDO MdBC cheese; (6) Venezuelan buffalo milk. Panel (**A**), Coomassie Blue G-250 staining. Panel (**B**) and (**C**), immunoblotting with polyclonal antibodies against α_s1_-CN (1-22) and β-CN (195-209), respectively.

**Figure 2 foods-14-01193-f002:**
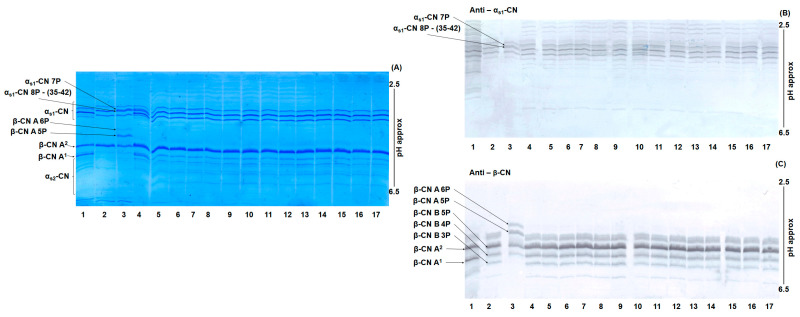
UTLIEF on a polyacrylamide gel (pH gradient 2.5–6.5) of mozzarella cheese samples. Lanes from left to right: (1) bovine casein standard; (2) Mediterranean buffalo casein standard; (3) foreign buffalo casein standard; (4–17) commercial PDO MdBC cheese samples. Panel (**A**) Coomassie Blue staining. Panel (**B**) and (**C**) immunoblotting with polyclonal antibodies against α_s1_-CN (1-22) and β-CN (195-209), respectively.

**Figure 3 foods-14-01193-f003:**
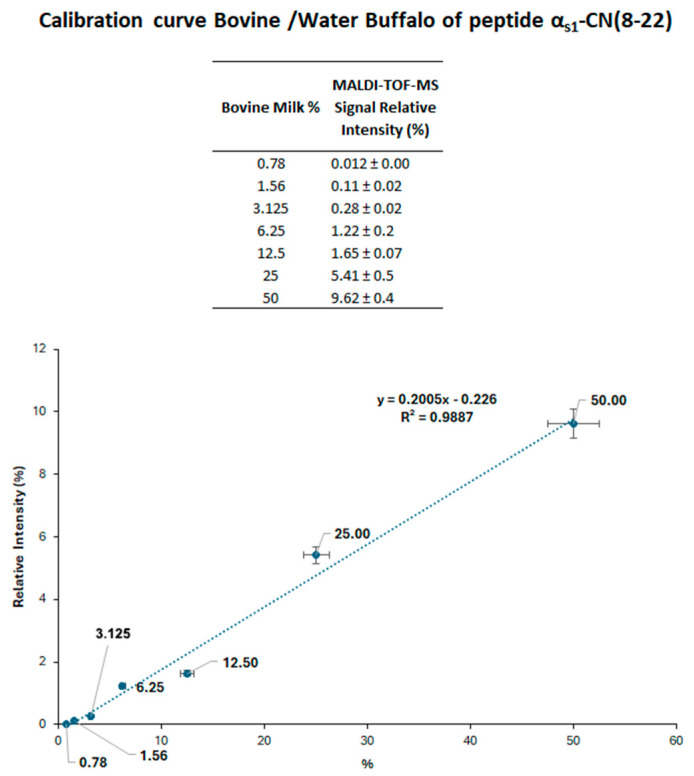
Calibration curve obtained from MALDI-TOF-MS analysis of tryptic α_s1_-CN (8–22) signature peptides in buffalo cheese samples containing known proportions of cow milk. Each point represents technical triplicates, with MALDI-TOF-MS signal intensities reported as mean ± standard deviation. Statistical significance was assessed using one-way ANOVA followed by Tukey’s HSD test (*p* < 0.05).

**Figure 4 foods-14-01193-f004:**
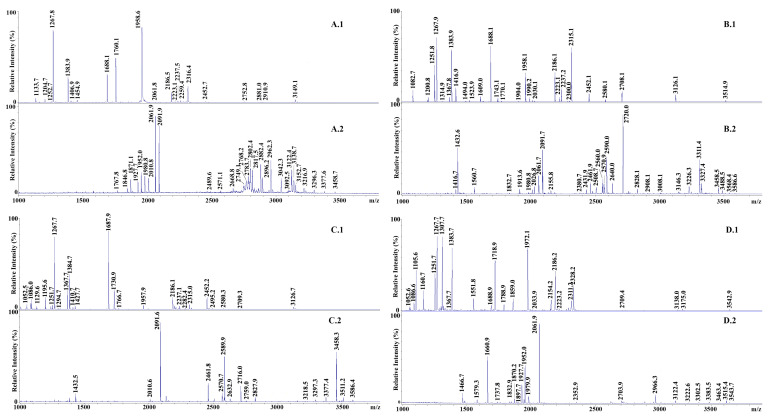
MALDI-TOF-MS analysis of MdBC cheese samples. Panel (**A.1**,**A.2**) IM22 sample. Panel (**B.1**,**B.2**) IM8 sample. Panel (**C.1**,**C.2**) foreign buffalo casein reference after tryptic digestion. Panel (**D.1**,**D.2**) ovine casein reference after tryptic digestion. Numbering 1 and 2 refers to samples before and following phosphopeptide enrichment.

**Figure 5 foods-14-01193-f005:**
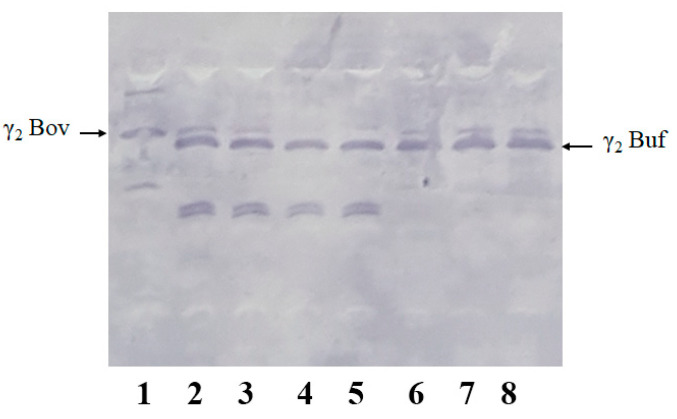
UTLIEF of casein plasminolysate combined with immunoblotting using polyclonal antibodies against β-CN (106–110) allows detecting bovine and buffalo γ2-casein. Lanes: 1. Whole cow casein. 2–3. Commercial MdBC cheeses adulterated with bovine milk. 4. Sixty-five-day-old buffalo blue cheese. 5–8. Buffalo milk mixed with increasing concentrations of bovine milk (0.5%, 2%, 5%, and 10% *v/v*).

**Table 1 foods-14-01193-t001:** Tryptic proteotypic peptides of caseins from dairy milk species. Reported are amino acid substitutions, phosphorylation state, and theoretical molecular mass values. Underlined are amino acids variably phosphorylated. M, Mediterranean; F, Foreign.

Protein Name	Peptide Sequence	Species Origin	Amino Acid Substitution/Deletion	Number of Phosphorylations	Molecular Mass (MH^+^)
**α_s1_-CN**	8–22	Buffalo M			1687.9
		Bovine	E → G_14_		1759.9
		Ovine C	P → S_12_; E → P_13_		1718.9
	23–34	Bovine/Buffalo M			1384.7
		Buffalo F	V → M_31_		1416.7
	23–33	Ovine	G → R_33_		1307.7
	37–58	Buffalo M	T → K_42_	2P	2571.0
		Bovine		2P	2598.1
	35–58	Buffalo F	I → V_36_; del (35–42)P	2P	1913.7
	43–58	Bovine		2P	1927.7
				1P	1847.7
				0P	1767.8
		Ovine	T → I_49_; I → A_57_	2P	1896.7
				1P	1816.7
				0P	1736.7
	104–124	Buffalo M/F			2452.3
	104–119	Bovine	L → S_115_; Q → R_119_	1P	1952.0
	106–119			1P	1660.8
	133–151	Buffalo M			2315.2
		Bovine	E→Q_148_		2316.2
		Ovine	G → A_133_; E → Q_137_; E → Q_148_		2328.2
**β-CN**	1–28	Buffalo Med		4P	3458.5
				3P	3378.5
				2P	3298.5
				1P	3218.6
				0P	3137.6
	1–25	Bovine	H → R_25_	4P	3122.3
				3P	3042.3
				2P	2962.3
				1P	2882.4
				0P	2801.4
	1–28	Bovine		4P	3477.5
		Buffalo F	G → S_10_	4P	3488.5
				5P	3568.4
	1–29	Buffalo M		4P	3586.5
		Bovine	H → R_25_	4P	3605.6
	33–48	Buffalo M		1P	2091.8
				0P	2011.9
		Bovine/Ovine/Buffalo F	M → T_41_	1P	2061.8
				0P	1981.9
	49–68	Buffalo M			2237.2
		Bov/Buf F	K → N_68_		2223.2
		Ovine	T → A_55_; K → N_68_		2197.2
	114–132	Buffalo M/Bovine			2168.0
		Ovine	N → K_131_		2183.1
	133–169	Buffalo M			4168.2
		Bovine	P → H_148_;		4208.2
		Ovine	L → V_40_; S → P_167_		4165.3
**κ**-CN	35–68	Ovine	K → R_46_; A → V_65_		4065.1
	69–86	Buffalo M/F			1989.1
		Bovine	K → R_46_		1979.1
		Ovine	K → R_47_; I → T_73_		1947.1

**Table 2 foods-14-01193-t002:** Nano-LC-ESI-MS/MS-based identification of buffalo and bovine casein-derived peptides in the pH 4.6 water-soluble extract and the tryptic digest of the pH 4.6 insoluble fraction obtained from a 65-day-old buffalo blue cheese.

Cheese Fractions	Water-Soluble Extract			Trypsinolyzed Water-Insoluble Extract	
Species	Buffalo	Bovine		Buffalo	Bovine
Proteins	Peptide Number				
α_s1_-CN	611	14		17	3
β-CN	1268	Var A1	14	26	0
		Var A2	7		
α_s2_-CN	603	29		31	5
k-CN	346	1		6	3
Total peptides	2828	65		80	11

**Table 3 foods-14-01193-t003:** Mass signal intensity ratios of orthologous peptides from bovine and buffalo β-CN in the pH 4.6 water-soluble extract obtained from a 65-day-old buffalo blue cheese. Data were derived from Appendix A.

Species	Orthologous Peptides	Mass	Start	End	Intensity	Bovine β-CN A1/ Buffalo β-CN Ratio
sp|Q9TSI0|CASB_BUBBU	DKIHPFAQTQSLVYPFPGPI**PK**	2479.33	47	68	1.55 × 10^8^	0.01
sp|P02666|CASB_BOVIN_VAR_A1	DKIHPFAQTQSLVYPFPGPI**HN**	2505.28	47	68	1.86 × 10^6^	
sp|Q9TSI0|CASB_BUBBU	SLVYPFPGPI**PK**SLPQ	1738.97	57	72	1.30 × 10^9^	0.01
sp|P02666|CASB_BOVIN_VAR_A1	SLVYPFPGPI**HN**SLPQ	1764.92	57	72	6.59 × 10^6^	
sp|Q9TSI0|CASB_BUBBU	YPFPGPI**PK**SLPQ	1439.78	60	72	9.87 × 10^8^	0.01
sp|P02666|CASB_BOVIN_VAR_A1	YPFPGPI**HN**SLPQ	1465.74	60	72	7.65 × 10^6^	
sp|Q9TSI0|CASB_BUBBU	SLVYPFPGPI**PK**	1313.74	57	68	4.00 × 10^8^	0.03
sp|P02666|CASB_BOVIN_VAR_A1	SLVYPFPGPI**HN**	1339.69	57	68	1.18 × 10^7^	
sp|Q9TSI0|CASB_BUBBU	FPGPI**PK**SLPQ	1179.67	62	72	1.79 × 10^8^	0.03
sp|P02666|CASB_BOVIN_VAR_A1	FPGPI**HN**SLPQ	1205.62	62	72	5.97 × 10^6^	
sp|Q9TSI0|CASB_BUBBU	LVYPFPGPI**PK**	1226.71	58	68	1.51 × 10^8^	0.04
sp|P02666|CASB_BOVIN_VAR_A1	LVYPFPGPI**HN**	1252.66	58	68	5.92 × 10^6^	
sp|Q9TSI0|CASB_BUBBU	FPGPI**PK**	754.44	62	68	5.89 × 10^7^	0.15
sp|P02666|CASB_BOVIN_VAR_A1	FPGPI**HN**	780.39	62	68	8.71 × 10^6^	
sp|Q9TSI0|CASB_BUBBU	GPI**PK**SLPQ	935.54	64	72	5.74 × 10^8^	0.07
sp|P02666|CASB_BOVIN_VAR_A1	GPI**HN**SLPQ	961.50	64	72	3.97 × 10^7^	
sp|Q9TSI0|CASB_BUBBU	HKEMPFPKYPVEP**F**	1744.86	106	119	4.94 × 10^8^	0.05
sp|P02666|CASB_BOVIN	HKEMPFPKYPVEP**L**	1710.88	106	119	2.32 × 10^7^	
sp|Q9TSI0|CASB_BUBBU	MPFPKYPVEP**F**	1350.67	109	119	3.31 × 10^8^	0.04
sp|P02666|CASB_BOVIN_VAR1	MPFPKYPVEP**L**	1316.68	109	119	1.36 × 10^7^	
sp|Q9TSI0|CASB_BUBBU	FPKYPVEP**F**	1122.58	111	119	1.81 × 10^9^	0.05
sp|P02666|CASB_BOVIN_VAR1	FPKYPVEP**L**	1088.59	111	119	9.28 × 10^7^	
sp|Q9TSI0|CASB_BUBBU	YPVEP**F**	750.36	114	119	3.68 × 10^8^	0.02
sp|P02666|CASB_BOVIN	YPVEP**L**	716.37	114	119	5.65 × 10^6^	
					Average value	0.04

**Table 4 foods-14-01193-t004:** Comparative summary of analytical methods used to detect MdBC cheese adulteration.

Analytical Method	Target Analyte	Sensitivity	Species Differentiation	Quantification Capability	Time Required	Suitability for Routine Testing
Gel Isoelectric Focusing (IEF)	γ-Caseins (plasmin digestion)	~0.5% cow milk	Cannot differentiate Mediterranean vs. foreign buffalo	No	~3 h	Limited (requires skilled personnel)
UTLIEF Electrophoresis	Intact caseins and hydrolyzed peptides	~1% cow milk	No	No	~3–4 h	Moderate
MALDI-TOF-MS	Proteotypic peptides from caseins	~0.1–0.5% cow milk	Yes (bovine, ovine, caprine, buffalo)	Semi-quantitative	~2 h	High in centralized labs
nano-LC-ESI-MS/MS	Casein-derived peptides (proteomic fingerprinting)	<0.1% cow milk	Yes (species-specific markers)	Yes (accurate quantification)	~4–6 h	Gold standard for official testing
Phospho-proteomics	Phosphorylation variants of caseins	~0.5% cow milk	Yes	Yes	~5–7 h	Advanced research applications
HPLC (Italian Official Method)	Bovine β-Lactoglobulin A	~0.5–1% cow milk	Yes (bovine-specific)	Yes	~2–3 h	Used for official routine analysis
Immuno-chromatographic Strip Test	Bovine IgGs in milk	~1% cow milk	No (only cow detection)	No	~10 min	Excellent for rapid on-site screening

## Data Availability

The original contributions presented in this study are included in the article/Appendix A. Further inquiries can be directed to the corresponding author.

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
