# Peer review of "Integrated Gel Electrophoresis and Mass Spectrometry Approach for Detecting and Quantifying Extraneous Milk in Protected Designation of Origin Buffalo Mozzarella Cheese"

_foods, 2025, doi:10.3390/foods14071193_

Round 1

Reviewer 1 Report

Comments and Suggestions for Authors

I carefully read this work A Comparative Study Using Gel Electrophoretic and Mass Spectrometric to Detecting and Quantifying Non-Buffalo Milk in 'Mediterranean Buffalo' Mozzarella Cheese. I feel that this is a significant study that provides a feasible method for distinguishing the authenticity of dairy products. The authors are advised to make the following revisions:

  1. Would replacing “Comparative” with “Combined” in the title be more suitable for the theme of this paper?
  2. The abstract contains too many unnecessary expressions. The authors should rewrite it according to the structure of background, methods, and results.
  3. The introduction should be presented in paragraphs.
  4. The purity of the reagents used should be supplemented in the Materials and Methods section.
  5. Line 115, the unit of time should be changed to “min”.
  6. Line 200, the origin and country of the Maldi-Tof-MS should be added.
  7. Where is the Marker in Figure 1? Please supplement it. The quality of Figures B and C is too poor; the authors should repeat the experiments and upload new images.
  8. In Figure 2, what do the numbers 1-22 represent? Where is the Marker? Figure B is not clear. Figure C shows some bands with diffusion. The authors should repeat the experiments for Figures B and C and upload new images.
  9. All tables in the manuscript should be formatted using the “three-line” system.

     10. Table 1 is too disorganized and should be redrawn.

Author Response

I carefully read this work A Comparative Study Using Gel Electrophoretic and Mass Spectrometric to Detecting and Quantifying Non-Buffalo Milk in 'Mediterranean Buffalo' Mozzarella Cheese. I feel that this is a significant study that provides a feasible method for distinguishing the authenticity of dairy products. The authors are advised to make the following revisions:

          1          Would replacing “Comparative” with “Combined” in the title be more suitable for the theme of this paper?

Answer. The title was revised by replacing the term "Comparative" with "Integrated Approach" to emphasize the complementary integration of gel electrophoresis and mass spectrometry techniques.

          2          The abstract contains too many unnecessary expressions. The authors should rewrite it according to the structure of background, methods, and results.

Answer. The Abstract has been thoroughly revised according to your recommendation, clearly following a structured sequence of background, methods, and results. However, we retained the single-paragraph format to maintain consistency with the common formatting practice of numerous articles previously published in Foods.

          3          The introduction should be presented in paragraphs.

Answer. The introduction section was reorganized into clearly defined paragraphs, redundant phrases were eliminated, and linguistic complexity was clarified while maintaining scientific rigor. It was noted to the editor that typically introductions in this journal are not divided into paragraphs, leaving the final formatting decision to the editor.

          4          The purity of the reagents used should be supplemented in the Materials and Methods section.

Answer. We have explicitly included the purity details of all reagents utilized within the Materials and Methods section, improving methodological transparency without compromising readability. This format aligns with established practices observed in Foods.

          5          Line 115, the unit of time should be changed to “min”.

Answer. The unit for time on line 115 has been corrected to "min," in accordance with your suggestion.

          6          Line 200, the origin and country of the Maldi-Tof-MS should be added.

Answer. We have added the origin and country of manufacture for the MALDI-TOF-MS instrument, as requested (line 200), to provide comprehensive methodological details.

          7          Where is the Marker in Figure 1? Please supplement it. The quality of Figures B and C is too poor; the authors should repeat the experiments and upload new images.

Answer. We acknowledge your concern regarding markers and image clarity. Figures 1B and 1C represent western blot experiments, inherently prone to slight protein diffusion during transfer onto membranes. Such minor smearing is typical and extensively documented in scientific literature for these assays. Additionally, the absence of markers is intentional. The aim of these experiments was comparative, based on the isoelectric point, to distinguish the presence of foreign buffalo milk in mozzarella. This is characterized by variant A, which has 5 and 6 phosphate groups with isoelectric points that differ from the other Mediterranean variant B, which has 4 phosphate groups. Nonetheless, figures have been revised, and enhanced versions with improved clarity and format are include.

          8          In Figure 2, what do the numbers 1-22 represent? Where is the Marker? Figure B is not clear. Figure C shows some bands with diffusion. The authors should repeat the experiments for Figures B and C and upload new images.

Answer. Numbers indicated (1–22 and 195–209) correspond to protein sequence regions recognized by the respective polyclonal antibodies. Regarding concerns about clarity and marker presence, we refer you to our detailed response provided for Figure 1. 

          9          All tables in the manuscript should be formatted using the “three-line” system.

Answer. All manuscript tables have been reformatted using the standardized "three-line" table format, ensuring consistency and professional appearance.

        10         Table 1 is too disorganized and should be redrawn. 

Answer. Table 1 contains detailed and complex data regarding protein sequence variations among species. Although this data format is standard and commonly used in the field, we recognize your concern about readability. Consequently, we reorganized Table 1 into a single-column layout to enhance clarity and accessibility to readers.

We sincerely appreciate your suggestions and hope these revisions fully address your concerns, significantly improving the manuscript's overall quality.

Reviewer 2 Report

Comments and Suggestions for Authors

The article entitled “A Comparative Study Using Gel Electrophoretic and Mass Spectrometric to Detecting and Quantifying Non-Buffalo Milk in 'Mediterranean Buffalo' Mozzarella Cheese” presents an interesting subject and actual theme. Briefly, the article investigates methods for detecting falsification of Mozzarella di Bufala Campana (MdBC), a Protected Designation of Origin (PDO) cheese made exclusively from Mediterranean buffalo milk. Given the risk of falsifications by additions of non-buffalo or foreign buffalo milk, the researchers developed a multi-platform analytical strategy combining three techniques as follows: the detection of foreign casein using Gel Electrophoresis and Immunoblotting; Mass Spectrometry Techniques like MALDI-TOF-MS and nano-LC-ESI-MS/MS and last but not least, Peptidomics & Phosphoproteomics which Identifies species and breed-specific peptide markers to detect non-PDO milk.

First of all I can notice from the very beginning a few spelling and writing mistakes. For example:

L20: „Here we presents...” must become “Here we present”.

L23, The Title: “Gel electrophoretic” in my opinion must become “Gel electrophoresis”.

L29: “confirming extraneous milk proteins” – “confirming the presence of extraneous milk proteins”.

L59: “IEF does not to differentiate” must become “IEF does not differentiate”.

L88: “we seeks to improve” – “we seek to improve”.

There are others mistakes (e.g 436 these peptides escaping detection - these peptides escaped detection ) so please be careful regarding one more time check of the manuscript.

L91: “Materials and Methods” – I read very carefully this section but unfortunately I could not identify the “statistical part” for this article. Also this very important interpretation is missing from the “Discussion” section to. I would like to let you know that any research that wants to have a real impact in the scientific community must be validated by data and statistical interpretation. So it’s mandatory to include this part into your manuscript. A few aspects to consider and to guide you regarded this part: you can perform statistical tests to confirm the significance of differences in casein profiles between species etc.

L161: “Calibration Curves” – Please try to give more information about the calibration curves. It is not entirely clear… For example how the calibration curves were validated for different milk species? The answer should appear into the main manuscript.

L295, 301, 353: The titles for the figures and tables are not good. These titles give the impression that they are actually interpretations from the results chapter and not the actual titles of the tables or figures. I suggest that you try to adapt the titles and make them more specific for figures and tables.

I’m wondering, could those detection approach be adapted for real-time or rapid screening of buffalo mozzarella at production sites? I think that this will be very interesting and helpful for the impact of the present research. Please include your reasoned answer into the main manuscript.

Please try to follow my suggestions regarded the form of the article as outlined bellow:

The abstract is very technical and to complex. This is good for the main body of the article but the abstract section should be much simpler and concise. l Consider simplifying some terms so a broader audience can read the key findings.

Into the “Results” section consider presenting a table summarizing key detection results across different methods.

I strongly recommend one more time the English revision of the present manuscript and please revise the reference list according to the journal guidelines (some references appear more than once, possibly due to an editing error; some references use first names instead of initials; missing or incorrect volume, issue, and page numbers).

In the end I want to congratulate the authors for their work and I am looking forward for their response.

Comments on the Quality of English Language

Dear authors, 

I have identified several errors of writing and spelling mistakes in your manuscript. Some of them I have listed in the review, but there are more. Please try to revise English once more.

Thank you!

Author Response

Authors' Response to Reviewer #2:

Thank you for your thorough review and valuable comments, which have significantly helped us enhance the quality and clarity of our manuscript. Below, we provide detailed responses to each of your points:

First of all, I can notice from the very beginning a few spelling and writing mistakes.

For example:

L20: „Here we presents...” must become “Here we present”.

L23, The Title: “Gel electrophoretic” in my opinion must become “Gel electrophoresis”.

L29: “confirming extraneous milk proteins” – “confirming the presence of extraneous milk proteins”.

L59: “IEF does not to differentiate” must become “IEF does not differentiate”.

L88: “we seeks to improve” – “we seek to improve”.

There are others mistakes (e.g 436 these peptides escaping detection - these peptides escaped detection ) so please be careful regarding one more time check of the manuscript.

Answer. We acknowledge your comment regarding several grammatical and spelling errors. The entire manuscript has undergone rigorous proofreading by a native English speaker, and all errors, including the specific examples provided, have been corrected. We greatly appreciate your careful review, which allowed us to improve the manuscript's readability and precision.

Specifically:

  • L20: Corrected from "Here we presents" to "Here we present."
  • L23 (Title): Revised from "Gel electrophoretic" to "Gel electrophoresis."
  • L29: Corrected from "confirming extraneous milk proteins" to "confirming the presence of extraneous milk proteins."
  • L59: Corrected from "IEF does not to differentiate" to "IEF does not differentiate."
  • L88: Corrected from "we seeks to improve" to "we seek to improve."
  • L436: Corrected from "these peptides escaping detection" to "these peptides escaped detection."

L91: “Materials and Methods” – I read very carefully this section but unfortunately I could not identify the “statistical part” for this article. Also this very important interpretation is missing from the “Discussion” section to. I would like to let you know that any research that wants to have a real impact in the scientific community must be validated by data and statistical interpretation. So it’s mandatory to include this part into your manuscript. A few aspects to consider and to guide you regarded this part: you can perform statistical tests to confirm the significance of differences in casein profiles between species etc.

Answer. In response to your valuable suggestions, we have improved the section “2.11. Statistical analysis” by clearly indicating the statistical methods used, including the justification for their selection. We have specified that we employed one-way ANOVA followed by Tukey’s HSD test to confirm significant differences in casein peptide profiles between species in technical triplicate analyses. Additionally, we have ensured that our statistical results are explicitly cited and discussed in the “Discussion” section, thus strengthening the rigor and scientific impact of the manuscript.

L161: “Calibration Curves” – Please try to give more information about the calibration curves. It is not entirely clear… For example how the calibration curves were validated for different milk species? The answer should appear into the main manuscript.

Answer. Authors’ response: Thank you for pointing out the need for clarification regarding the calibration curves. We have provided additional information in the revised manuscript (lines 780–787) to clearly explain the rationale behind the choice and validation of the peptides employed for calibration. Specifically, the calibration curves were intended to quantify contamination levels of bovine milk in Mozzarella di Bufala Campana cheese. We initially evaluated several potential proteotypic peptides (e.g., β-CN(33-48)1P and β-CN(1-25)4P) to distinguish bovine milk contamination. However, these peptides were found to be unsuitable for quantitative purposes due to inherent variability or instability. Their unsuitability is primarily attributable to natural heterogeneity caused by the natural coexistence of phosphorylated and partially phosphorylated isoforms, variability possibly introduced by milk processing (thermized, pasteurized), phosphatase activity affecting phosphorylation status, and unpredictable partial oxidation of methionine residues during thermal treatment or cheesemaking processes. As clearly explained in paragraph 3.3.4 ("Calibration and Quantitative Accuracy"), despite these limitations in quantification, the presence of these peptides still provides unequivocal evidence of bovine milk contamination. For ovine species identification, we selected the peptide αs1-CN(8-22) (1718.9 Da), which does not present issues of variable phosphorylation or oxidation due to the absence of modifiable residues. Thus, this peptide allowed robust quantification of ovine contamination. Furthermore, to better illustrate our quantification methodology, we have detailed in paragraph 3.3.3 (“Quantification of Adulteration in Buffalo Mozzarella Cheese”) that we adopted a simplified approach based on the calculation of relative signal intensity ratios of orthologous proteotypic peptide pairs from different species. This method relies on the assumption that peptide ion signal intensities are proportional to their parent protein abundances and that amino acid substitutions without net charge changes do not significantly affect ionization efficiency. We verified the validity of this strategy through cheeses obtained from controlled mixtures of buffalo and cow milk ranging from 0.78% to 50% cow milk. Additionally, the peptide intensity ratio approach was preferred for practical considerations: avoiding the complexities and costs associated with synthetic peptide standards, maintaining methodological simplicity, and ensuring reliability within the tested contamination range. We believe this clarification effectively addresses your concerns. Additionally, we have included this comprehensive explanation within the main manuscript to enhance clarity and scientific rigor.

L295, 301, 353: The titles for the figures and tables are not good. These titles give the impression that they are actually interpretations from the results chapter and not the actual titles of the tables or figures. I suggest that you try to adapt the titles and make them more specific for figures and tables.

Answer. Although we initially considered our titles appropriate, we revised the titles of Figures 1 and 2 and Table 1 according to your feedback, ensuring they better reflect their content and purpose while enhancing clarity.

I’m wondering, could those detection approach be adapted for real-time or rapid screening of buffalo mozzarella at production sites? I think that this will be very interesting and helpful for the impact of the present research. Please include your reasoned answer into the main manuscript.

Answer. Your insightful suggestion about the potential for real-time or rapid screening at production sites has been addressed explicitly in the revised Conclusions section. Following your recommendation, we have included a new Table (Table 6) within the "Results" section, summarizing key detection outcomes obtained from the different analytical methods employed. Additionally, we introduced paragraph 3.11 ("Comparative Analysis of Detection Methods") to discuss clearly the advantages and limitations of each analytical strategy, emphasizing the strength of combining multiple methods for robust detection and traceability in Mozzarella di Bufala Campana authenticity assessments.

Please try to follow my suggestions regarded the form of the article as outlined bellow:

The abstract is very technical and to complex. This is good for the main body of the article but the abstract section should be much simpler and concise. l Consider simplifying some terms so a broader audience can read the key findings.

Answer. The Abstract has been fully revised and simplified in line with your suggestion. It now clearly communicates key findings and methods in a concise and accessible manner, suitable for a broader readership.

Into the “Results” section consider presenting a table summarizing key detection results across different methods.

Answer. We thank you for recommending the inclusion of a summary table of key detection results. We have introduced Table 6 in the "Results" section and paragraph 3.11 ("Comparative Analysis of Detection Methods") to systematically summarize and compare the analytical approaches employed. This addition enhances the manuscript’s clarity and utility.

In the end I want to congratulate the authors for their work and I am looking forward for their response.

Answer. The Authors thank the Reviewer for this comment.

Comments on the Quality of English Language

Dear authors, 

I have identified several errors of writing and spelling mistakes in your manuscript. Some of them I have listed in the review, but there are more. Please try to revise English once more.

Answer. We sincerely appreciate your thoughtful feedback and supportive comments, and we hope our revisions adequately address all your concerns and suggestions.

Round 2

Reviewer 1 Report

Comments and Suggestions for Authors

The manuscript could be published in present form.

Reviewer 2 Report

Comments and Suggestions for Authors Dear Authors, Thank you very much for responding to all of my queries and for make the appropriate modifications to the manuscript as I previously suggested.  I have no further queries.  Thank you and wish you good luck in the future research projects!